# The Effect of Dust Storm on Sea Surface Temperature in the Western Basin of Persian Gulf

Masoud Torabi Azad [1], Kamran Lari [1], Rana Oudi [2], Tayeb Sadeghifar [3,4] and Ozgur Kisi [5,6,*]

1 Faculty of Marine Science and Technology, Islamic Azad University, North Tehran Branch, Tehran 14515-775, Iran; m_azad@iau-tnb.ac.ir (M.T.A.); k_lari@iau-tnb.ac.ir (K.L.)
2 Physical Oceanography, Islamic Azad University, North Tehran Branch, Tehran 14515-775, Iran; ra62na@yahoo.com
3 Department of Marine Physics, Faculty of Marine Sciences, Tarbiat Modares University, Tehran 14115-336, Iran; t.sadeghifar@modares.ac.ir or tsadeghifar90@gmail.com
4 Department Physics, Technical and Vocational University (TU), Tehran 16846-13114, Iran
5 Department of Civil Engineering, University of Applied Sciences, 23562 Lübeck, Germany
6 Department of Civil Engineering, Ilia State University, 0162 Tbilisi, Georgia
* Correspondence: ozgur.kisi@th-luebeck.de

**Abstract:** A dust storm is one of the costliest and most destructive events in many desert regions. This research investigates the effect of dust storm on sea surface temperature (SST) in the western zone of the Persian Gulf, especially Bushehr Province and its beaches in the years 2008 and 2009. Therefore, some climate and sea parameters such as SST, salinity, air temperature, wind velocity and direction, evaporation, horizontal visibility, sunshine hours and radiation, simultaneously measured in a specific period of time, were analyzed by comparing each of them with satellite data. Sea surface temperature analysis in summer shows that the maximum SST in Persian Gulf along neighbor waters to Bushehr County and central regions in northern section of Persian Gulf is about 34–36 °C. The SST amplitude variation in these places in summer ranges from 28 to 34 °C and when there are dust phenomena, it is from 29.5 to 31 °C. The outcome of this study shows that the SST increases during dusting phenomena and this causes an increase in vapor and as a result a decrease in temperature occurs. On the other hand, vapor increase leads to a growth in the amount and layer of earth's cloud cover and finally it causes an effective decrease in short-wave sunshine and the temperature and the vapor on surface decrease. As a result, the decrease in sea surface temperature terminates.

**Keywords:** dust storm; sea surface temperature; evaporation; horizontal visibility; Persian Gulf

## 1. Introduction

The dust storm has been seen in Iran for several years, especially in the southern provinces of the country, including Bushehr and Khuzestan. This phenomenon, in addition to the emergence of heavy financial losses, has made life difficult for people and it also threatens health.

The situation of Bushehr province between 27°17′ and 50°8′ has made it one of the warmest regions in the country. Due to the geographical development of the province, the air masses of the south-west and west are able to enter the region and have special effects on the province's climate. The extent of the province, at about 3 degrees of latitude, makes significant changes in some climatic parameters such as sunny hours, exposure, radiation, and other energies received from the sun, each of which affects the climate conditions of the province. The province of Bushehr is one of the provinces which has the lowest rainfall in the country and the most important systems affecting it in winter are Mediterranean and Sudanese low pressures and northern and northwestern areas; other high-pressure centers also have a moderate effect on this province during the cold seasons of the year.

Bushehr province receives summer heat from the Sahara, Africa, and Iran's desert. During the summer, the low pressure of the Monsoon centers has limited effects on Bushehr.

Aridity from the Saudi, African, and Iranian deserts increases temperature and evaporation while decreasing rainfall in Bushehr [1]. Conditions in southern Iraq and northern Saudi Arabia cause the severe walnut phenomenon. The public is probably used to the dust & haze phenomenon. However, these suspended particles in the air are remnants of sandstorms, etc.

Desertification and green belt are being imposed on the passage of this phenomenon, especially in Khuzestan and the Iraqi border strip; this has led to an increase in green areas per capita, creating forest parks, designing, producing and distributing appropriate masks, decay and using oil buyers on the sand in Iraq and Iran, and Iran and the UAE to fund the destruction of deserts in Iraq, using political means on swimming to integrate the countries involved in this problem and co-operation among the countries, reduce unnecessary traffic in free space and return rivers to the main path are some of the necessary measures that could be used to reduce the dust effects [2].

Wang et al. (2007) investigated China's Doughnut Hurricane during 2002–2000 based on data provided by China's Meteorological Department and essential features, including the source, movement path and affected areas including the Chinese Sea [3]. Zhang and Chai (2008) investigated the Asian Dust Storms, and its effects on the weather, environment, and oceans. The origin, path, and dimension of these particles were investigated and its effect was analyzed [4]. Jamalizadeh (2008) predicted dust storms using the artificial neural networks technique in Zabul [5]. Moreover, dust storms were analyzed in this region. Badarinath and Kumar Kharol (2009) investigated the dust storm in 2008 and the changes resulting from it in the Arab Sea, as well as the vertical profile of suspended particles. For this purpose, field and satellite data were used in the area under study [6]. Ashrafi et al. (2014) simulated dust storms in Iran by using the Hybrid Single Particle Lagrangian Integrated Trajectory (HYSPLIT). The suspended particles have harmful effects on human health, environment, and economy. This pollutant may be emitted from natural, or human resources. The main ratio of natural particles to atmosphere is due to air erosion from dry and semi-dry regions at the world scale. The identification of the dust source and the path simulation, using numerical techniques, are the main goals of the study. In the HYSPLIT model, dust and path simulation has been used and two cases have been studied (in May and June 2010). HYSPLIT dust is based on the dust storm algorithm for use in desert lands. This method is used to estimate the sensitive points and routes [7]. The distribution and transport of dust that occurred in Northeast Asia from 28 March to 2 April 2012 was investigated by [8]. Data from suspended particles less than 10 μm (10 pm) near the surface and using light detection and range from Earth to 18 km have been used in the study [9]. Esmail et al. (2019) investigated the performance of this system in arid and semi-arid regions that are prone to frequent dust storms. The results showed that intense dust that made the band less visible. In addition, they found that system performance can be improved by using short sections or multi-hop systems. Insignificant results showed that the performance of the system under dense dust was improved compared to fog disturbances indicating that dust produces a much higher damping than fog. Therefore, dust can be considered as the final damage for FSO bonds [10].

According to the Singer and Ganor 2003, as a result of the dust storm, annual precipitation has gradually increased in the near coasts [11]. The origin, dimensions, and materials forming the particles have been investigated and its effect on sea water pollution has been investigated. In the study by Ichoku et al. (2004), the experimental parameters, and the measured values of MODIS sensors on the Terra and diving satellite were continuously evaluated by ground-based measurements, which are used for different aerosol studies. These parameters provided a two-year loading and seasonal progressive assessments on several important regions on land and ocean-based on an average of 5 months [12–15].

Air-sea interaction leads to many unique phenomena in the atmosphere and ocean. One of them is the dust in the air which decreases the amount of horizontal vision. As a result of dust in the air, the air temperature will increase. One of the effects of this temperature increment is the increase in evaporation on sea surface, and thus, the reduction

in surface temperature. The weather conditions and its variations depend on the ocean and sea conditions, especially sea-level temperatures [8]. Therefore, any discussion on the weather, whether inadvertently, lead to a prior recognition of the sea surface temperature (SST) situation. To this end, Iran's researchers have been tasked to study various aspects of this issue on the Iranian coast. Since few studies have been conducted on the effects of wind and other meteorological factors such as dust, rainfall, evaporation and pollutant materials on parameters such as sea WST (WST) and salinity in the south and southwest of Iran, the present study investigates the effect of dust phenomenon on the surface temperature of water in the western Persian Gulf basin and presents the necessary predictions for this phenomenon. The results of the present study were analyzed in detail. In general, the effects of atmospheric factors and their interaction with sea water have been investigated.

The interaction between air and sea causes many unique phenomena in the atmosphere and the ocean. One of these phenomena is dust in the air, which will reduce the horizontal visibility. Due to the presence of dust in the air, the air temperature will increase. One of the effects of this increase in temperature is an increase in the rate of evaporation from the sea level and therefore a decrease in the surface temperature of the water. The effects of atmospheric factors on the Persian Gulf climate, surface water temperature and wind current due to wind are quite noticeable. Atmospheric conditions and their changes depend on ocean and sea conditions, especially SST. Therefore, any discussion or study of the desired weather condition leads to prior knowledge of the SST status.

The effects of wind and other meteorological factors such as wind speed, precipitation, evaporation and pollutants on parameters such as surface water temperature, salinity, sea currents, sea surface instability, etc. in the southern and southwestern parts of Iran have rarely been studied [16,17]. This study tries to investigate the effect of climate change (dust phenomenon) on the physical parameters of seawater in the western basin of the Persian Gulf and also to provide the necessary predictions in this regard and analyze the results in detail. The main aim of this study is to provide a preliminary study on the effect of atmospheric factors on the seas and oceans, a case study of the effect of atmospheric changes (dust phenomenon) on the physical parameters of water in the western basin of the Persian Gulf.

## 2. Material and Methods

The studied region is the western basin of the Persian Gulf. The depth of this watershed in the western part is about 10–30 m. The beaches of Bushehr region and the western basin of the Persian Gulf have special importance regarding the location of the nuclear plant in Bushehr and the spread of possible pollution, and also the dust storm, which is the phenomenon in the southwest of the country. Certainly, it is valuable to have complete information about the temperature changes in this region. On the other hand, comparing field data with satellite data, which is the source of many projects and studies, will help reducing the probable errors (Figure 1).

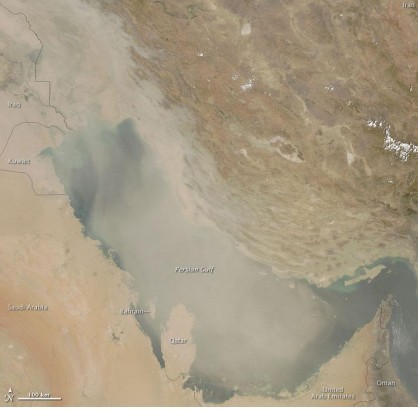 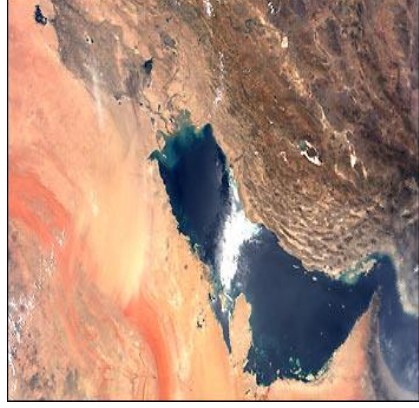

**Figure 1.** The ECO satellite image of Valdemar deployment in southwest Iran and the Persian Gulf.

Due to data limitations (as a result of low depth and low intensity of research patrols) in this basin and the high sea border of Bushehr province with the Persian Gulf, for water sampling and measuring, the related parameters measured at the Bushehr Sea synoptic station, the Directorate of Meteorology of this province, located 12 km from the coastal line, were used. The region's geographical location is such that the air masses from the South West and South are able to enter this region and leave special effects.

In this study, to investigate the effect of dust phenomenon on WST, the July 2008 and 2009 was considered. In August 2008, the dust scattered slightly in the sky, and in July 2009, the phenomenon of the walnut was maximum. In this study, during the two months of 2008 and 2009, diagrams of correlation, time series, frequency distribution, and WST contour lines were compared to show the changes in WST due to the dust phenomenon. Excel, SPSS, and Surfer software were used for statistical analysis, graphics, and contour lines.

## 3. Results

After data collection, field and satellite data were validated. For this purpose, the correlation of surface temperature with water and air temperature was first investigated through field and satellite measurements. To observe the scattering of field and satellite data, fitting lines and corresponding equations were obtained and the error rates were calculated [17]. The $R^2$ value was estimated as 0.9 which shows that there is a strong correlation between these two data. Therefore, a significant relationship was found between surface temperature (satellite) and field data, and the obtained errors between these data were found to be acceptable. There is a complete and direct correlation between field and satellite data. The same holds for air temperature value. After the surface temperature data modification of water and air temperature using the correlation equation, the surface temperature diagrams of spotting, air temperature, evaporation, radiation, and sunny watch were drawn. This is the most tangible effect of dust on the horizontal distance and the most significant of them were used for comparison. Based on data analysis, there was an indirect and part correlation between horizontal vision and WST.

The surface temperature of water is strongly correlated with the wind direction and parameters such as evaporation and sunshine hour and this is because the effects of several parameters on WST are dependent together and sometimes the effects of one of them on water surface eliminate another. To determine and compare surface temperature, air temperature, horizontal vision, speed and direction of wind, evaporation, radiation and sunny hour during the 2008 to 2009 years on the shores of Bushehr province, the related time series were drawn (Figures 2–7). The purpose of drawing such graphs is to identify the changes in these parameters during the months when there is dust in the air.

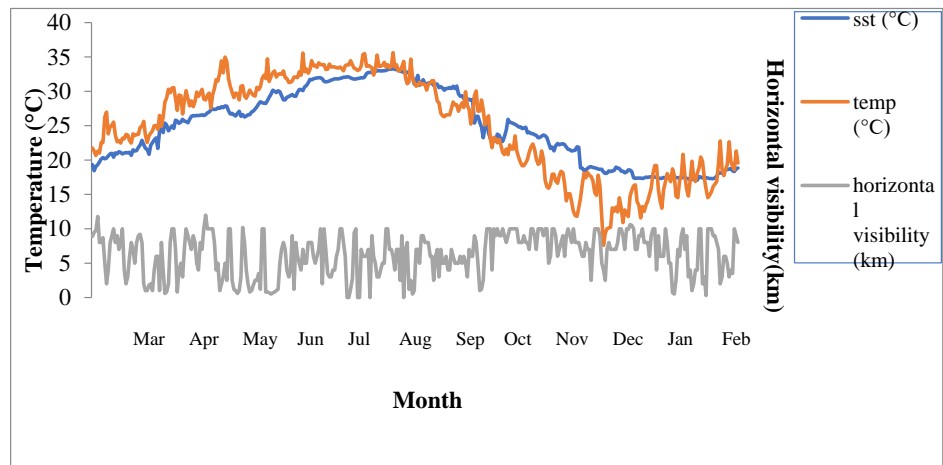

**Figure 2.** Time series of sea surface temperature, air temperature and horizontal vision in 2008 (sst: Sea surface temperature, temp: Air temperature).

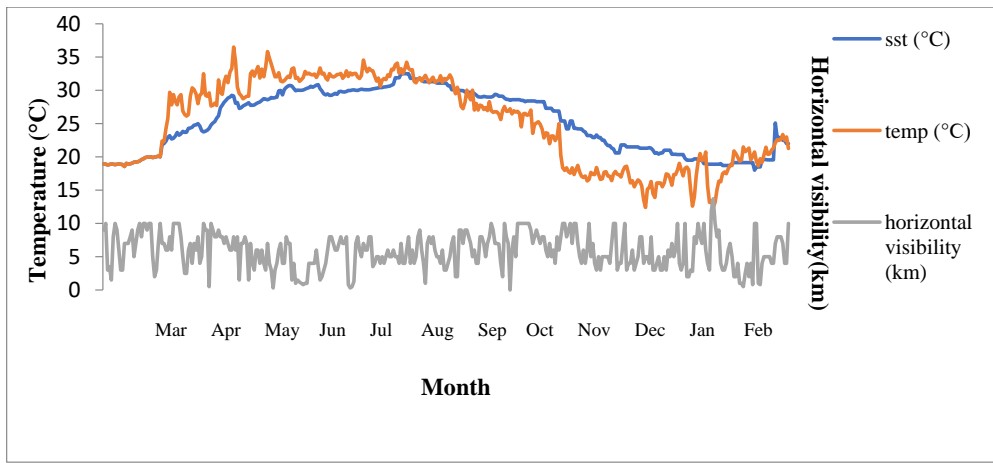

**Figure 3.** Time series of sea surface temperature, air temperature and horizontal vision in 2009 (sst: Sea surface temperature, temp: Air temperature).

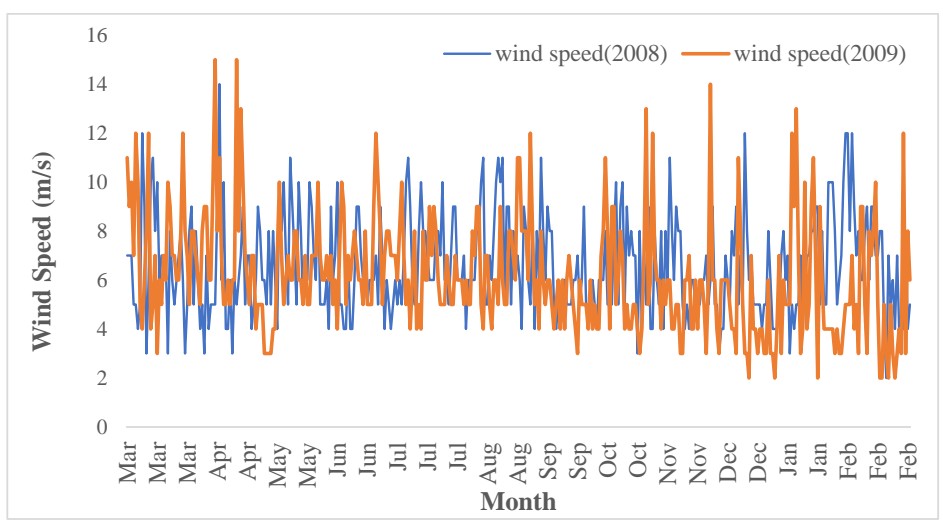

**Figure 4.** Wind speed time series in 2008 and 2009.

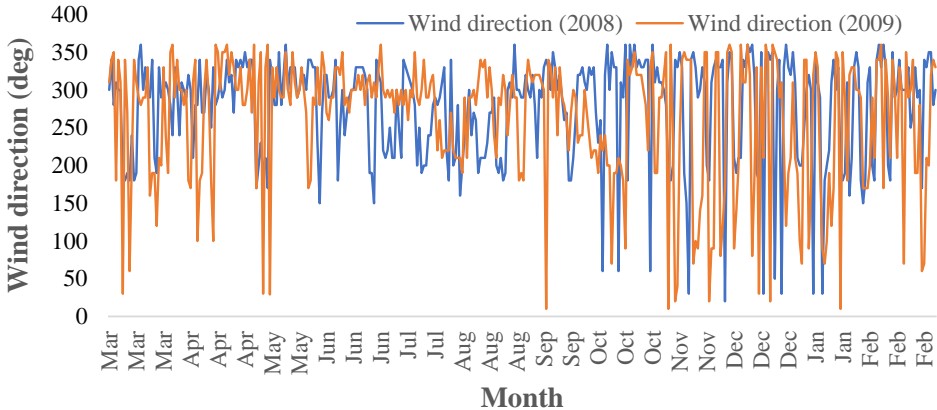

**Figure 5.** Wind direction series in 2008 and 2009.

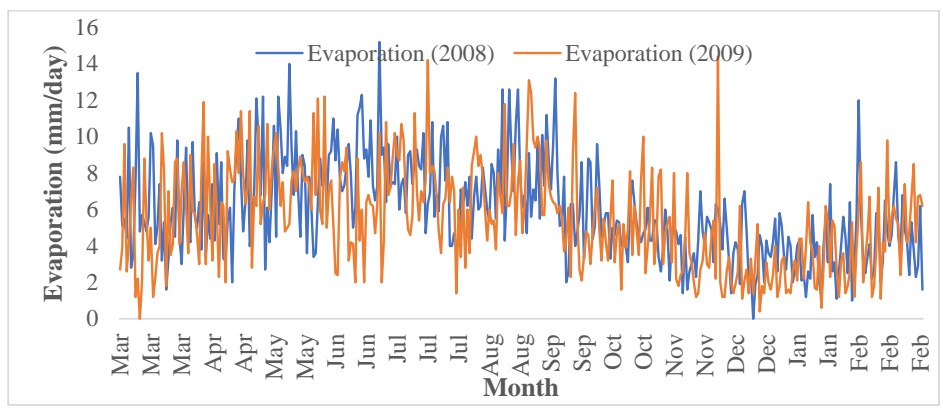

**Figure 6.** Evaporation time series in 2008 and 2009.

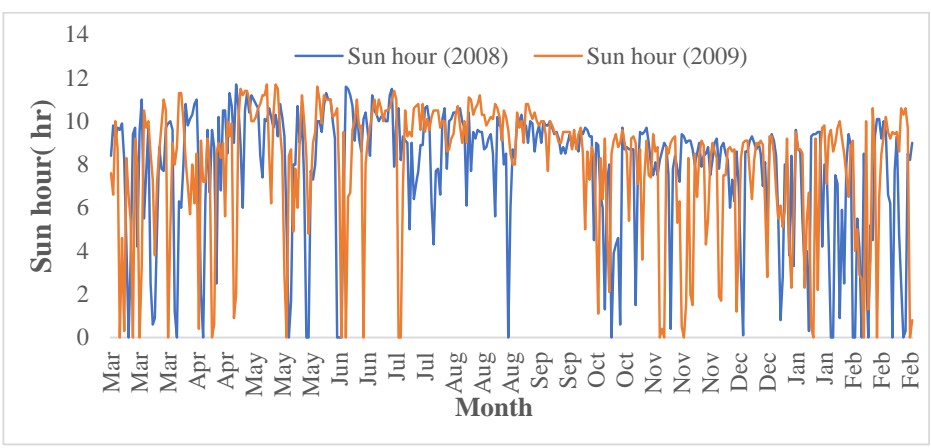

**Figure 7.** Time series of sunny hours in 2008 and 2009.

## 4. Discussion

According to Figures 2 and 3, air temperature is different from WST, and its variation is dependent on the direction of wind. If the temperature is lower than the WST, it would create instability. Cold weather on hot water causes instability. Stability would be achieved if the temperature is hotter than the WST. As shown in these charts, in the first half of the year 2008 and 2009, it was higher than the WST, while in the second half of the year, the surface temperature was higher than the air temperature. It is also clear that during the two rotary days, the temperature of air was increased slightly while the WST was decreased by decreasing the horizontal visibility distance.

In Figures 3–7, it is shown that with the increase in dust in the region in March 2008, sunny hours, temperature, evaporation are minimum, maximum and maximum, respectively. As evident from Figure 6, on the 31 March 2008 the sun arrived at minimum and evaporation increased. By comparing the results of March 2008, which was the beginning of the dust in 2008, it is shown that the highest surface temperature is 22.8 °C and the lowest surface temperature is 18.4 °C. Overall, in the spring 2008, the surface temperature of water increased. The scope of body temperature change is low, and it only decreased on rainy days. In July 2008, large dust was seen in the sky. On the 11th of July, horizontal vision decreased to 800 m. The temperature increased during the month with a range of changes from 30 to 35 degrees. Sunny hours also reached zero during the rainy days. The dominant wind has been in the northwest this month and southwest in many cases. The scope of the wind was high and between 4 and 12 m/s.

In early August in the southwest of the country, the dust phenomenon prevailed with winds, and the temperature increased to some extent due to the formation of summer conditions in the country. In the second half of the July, the country's southwestern coasts

were affected by dust from Iraq and Saudi Arabia, which led to a decrease in vision. Surface temperature changes in water are very low, and it only decreased during the dust days. Sunny times and evaporation were not as large and winds blowing were only observed in the early days of the month. In September 2008, the temperature and surface temperature of water is not very different, rather almost identical. The surface temperature of the water in September was decreasing. Sunny hours also have a lot of change, and by the end of the month when there was a lot of dust scattered in the air, the sunny hours reached zero.

Overall, during the summer of 2008, the surface temperature of water was increased, and after this season, the trend decreased. The range of surface temperature changes are between 17 and 35 °C and it slightly reduced during the cryogenic days. In 2008 October, climate change was not much different and it had a slight effect on WST. The dominant wind was in the northwest this month, and the extent of the sharp wind changes was high earlier this month. The surface temperature of water and air during this month had a decreasing rate. The slight difference between the surface and air temperatures caused the water instability. By this month, due to rainfall and cloud coverage, reduction rate and sunny hours decreased. The temperature of the water surface in December 2008 shows that the surface temperature is higher than the air temperature and that leads to severe water instability. Air temperature and WST during this month decreased and the changes in horizontal vision and sunny time occurred due to clouds in the sky, fog, and rain and rainfall. In the fall of 2008, the surface temperature of water decreased. The amplitude of WST changes are 20 to 32 °C and its amount is slightly reduced during the cryogenic days.

During the winter, the surface temperature of water decreased and after this season it increased. The amplitude of WST variations was 17–19 °C and its ratio decreased during the dusting days. According to the Figure 2, the surface temperature change increased in the first half of 2008 while it decreased in the second half. With the entry of the dust into this area, the sunny hours of March 2009 were low and the temperatures of the air and evaporation were maximum. Due to the amount of dust in the air during this month is low, the horizontal vision did not shrink.

Decreasing the sunny time in most sites justified the occurrence of dust phenomenon. The reduction in horizontal vision during these periods has low effect on WST. The trend of surface temperature change is steady and slow. In June 2009, we faced a noticeable decline in precipitation and temperature rise. The range of temperature changes during the month was 27–36 °C. The dust and winds over after this month intermittently spread throughout the southwestern part of the country. Horizontal vision was reduced. The surface temperature of water in June had a slow trend with the range of changes from 26 to 27 degrees Celsius. The wind direction was dominant in northwestern in this month.

Overall, during the spring of 2009, the surface temperature of water increased. The scope of the surface temperature change is low, and it only decreased in the rainy days. In July 2009, much dust appeared in the sky. The temperature increased during the month with a range of changes from 30 to 35 degrees. Sunny hours also reached zero during the rainy days. The dominant wind during this month was in the northwest and in the southwest in many cases. The range of the wind speed was high, between 4–12 m/s. In August 2009, the changes in WST were very low and had only a slight decrease during the dusty days. The changes in evaporation were minimal, and the sunny hours of the rainy days were even zero. The time series for water surface and air temperature in December 2009, shows that the WST is higher than the temperature of air. Weather temperature and water surface during this month decreased.

Overall, during the fall of 2009, the surface temperature of water decreased. The surface temperature changes for water surfaces were between 20 and 30 °C, and their volume decreased during the cryogenic days. During the winter of 2009, the surface temperature of water decreased and later in this season, it increased. The range of surface temperature changes is between 18 and 25 °C and it is slightly decreased during the rainy days. According to the Figure 8, the surface temperature change increased in the first half of 2009 while it decreased in the second half. The surface temperature changes in the water

were from 29.5 to 31 °C when the dust phenomenon occurred. During the occurrence of dust phenomenon, the air temperature changes were recorded at 30.5–32.5 °C (Figure 3). The range of horizontal vision changes is highly variable and during the rotary days, it decreased to 100 m. In the first half of 2008 and 2009, the temperature of water is higher than the surface temperature of the water, and in the second half of the year, the surface temperature of the water is higher than the temperature of the air. During the rotary period, the temperature of the air increased by the reduction in the horizontal sight distance, and the WST was slightly decreased.

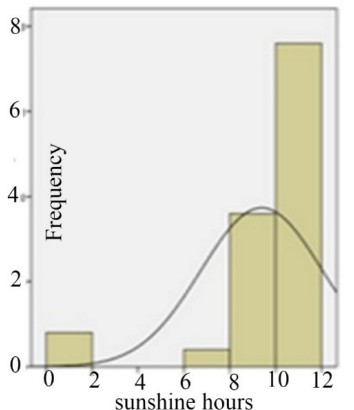 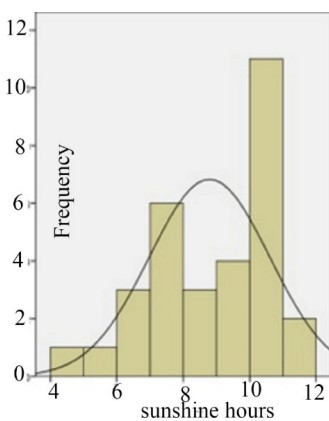

**Figure 8.** Sunshine hours' distributions in August 2008 (**Left**) and 2009 (**Right**).

Figures have also indicated seasonal changes in the WST in the Persian Gulf. Winter and summer differences of SST are about 15 °C. It is concluded that the summer surface temperature is due to the reduction in sun radiation, reduction in air temperature, wind speed, or risk of extinction. During peak days in 2008 and 2009, days of evaporation and sunshine decreased, and on some days, there was zero. Sunlight and sunshine hours, two interdependent factors have similar effects on the surface temperature of water and salinity in the region. By microscopic spectators, reduction in sun's radiation and sunshine hours during the months of May to September, the WST decreased. To compare the distribution of different parameters during different times, data frequency histogram was used (Figures 9–12). FΔ is called frequency bar width. As FΔ approaches to zero, ANOVA is converted to a continuous curve. The more data, the more likely it may occur. The sharpness of the corresponding histogram is shown that the dispersion of the parameter and its deviation is less than its average.

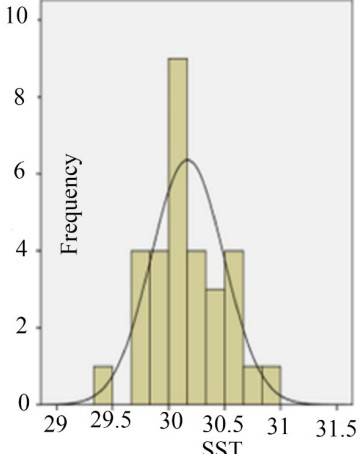 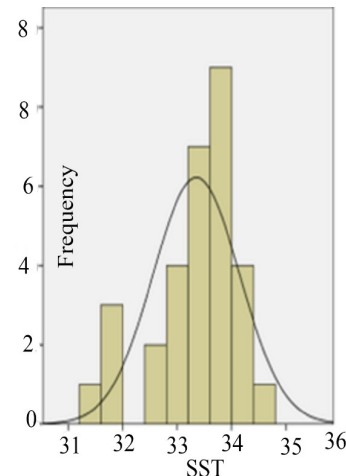

**Figure 9.** The frequency of the surface temperature of water in August 2008 (**Left**) and 2009 (**Right**).

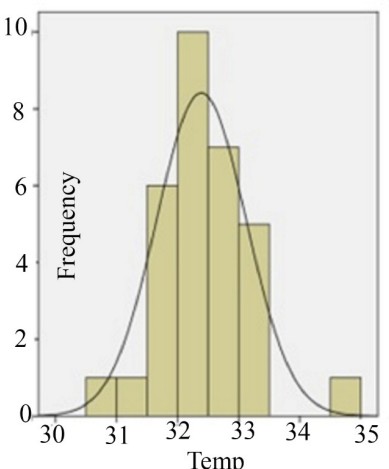
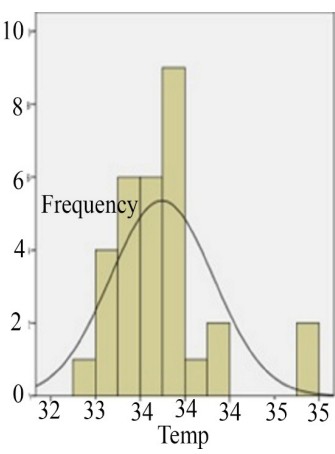

**Figure 10.** Temperature frequency in August 2008 (**Left**) and 2009 (**Right**).

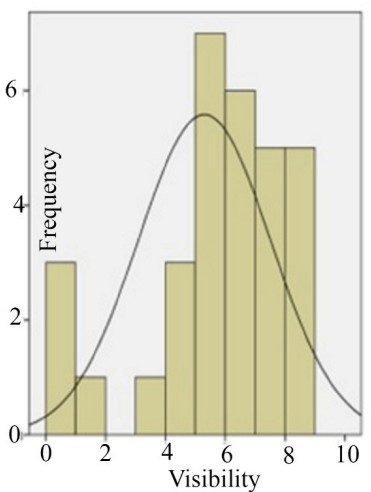
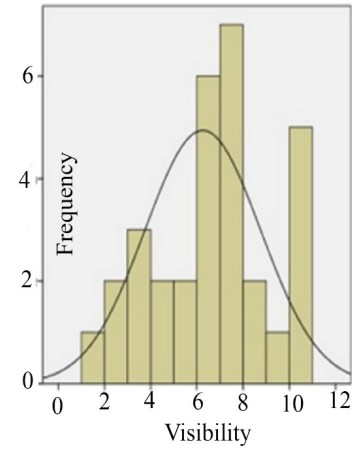

**Figure 11.** Temperature frequency in August 2008 (**Left**) and 2009 (**Right**).

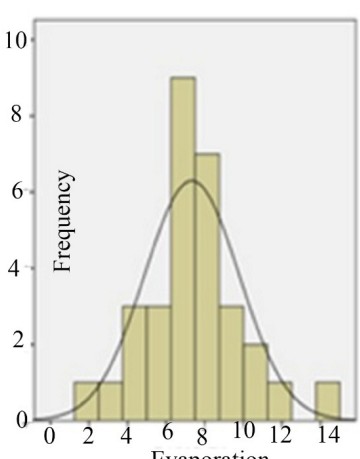
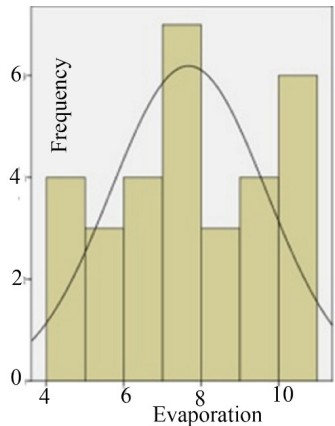

**Figure 12.** Evaporation frequency distribution in 2008 (**Left**) and 2009 (**Right**) August.

As shown in Figure 8, the highest measured values for WST in August 2008 were between 31.9 and 32 °C and in August 2009 they were between 30.5 and 30 °C. According to the Figure 9, the maximum values measured for air temperature in August 2008 range 33.5–34 °C and in 2009, they range 31–33 °C in the same month.

The highest values for evaporation were between 8 and 7 mm in August 2008 and 6–8 mm in August 2009. The highest measurements for the sunny hour in August 2008 were between 11 and 7 h and between 12 and 8 h in August 2009. The highest surface temperature of water for all months in July and August in 2008 and 2009 and August had the lowest WST (Figure 8).

The amount of horizontal viewing interval in the studied years is very variable and has the least visibility in August and September. Of course, in January and February, there was also a drastic decrease in the view of rainfall and moisture and fog in barley. According to the statistics of the stations, it was identified that December and January had the lowest amount of dust while the months of June and July had the highest amount of dust in the study area. The measured values for evaporation in June and July have the highest amount and the lowest amount was observed in January and February. The measured values are variable for sunny hours during 2008 and 2009, and have been very low in the days of dust and even reached zero. The strongest winds in February and March have the highest frequency compared to December and January as well as April. The dominant wind direction in these months is the beginning of a micro-influx to the country, with an approximate angle of 310 degrees or North West. However, the dominant wind direction occurred in December with an approx. angle of 200 degrees. The dominant wind direction in April is approximately 180 degrees. The wind of the north and northwest is a stable and steady in August and September. The north wind in the summer is the adornment of winter. Table 1 shows the horizontal and downwind rate and direction of the winds in August 2008 and 2009, during the days when the dust is recorded.

**Table 1.** The rate of horizontal and wind direction in August 2008 (high) and 2009 (lower), during days that dust is recorded.

| Date | Minimum Views (m) | The Dominant Wind | Wind Speed (m/s) |
| --- | --- | --- | --- |
| 28 July 2008 | 2000 | Northwest | 10 |
| 31 July 2008 | 2500 | Northwest | 6 |
| 1 August 2008 | 3000 | Northwest | 5 |
| 2 August 2008 | 3000 | Southwest | 5 |
| 3 August 2008 | 4000 | Southwest | 8 |
| 13 August 2008 | 1200 | Northwest | 7 |
| 14 August 2008 | 2800 | Northwest | 10 |
| 20 August 2008 | 6000 | Southwest | 9 |
| 29 July 2009 | 800 | Southwest | 5 |
| 30 July 2009 | 300 | Northwest | 6 |
| 31 July 2009 | 500 | Northwest | 4 |
| 1 August 2009 | 1300 | Northwest | 6 |
| 11 August 2009 | 3500 | Northwest | 9 |
| 12 August 2009 | 4000 | Northwest | 8 |
| 13 August 2009 | 5000 | Northwest | 6 |

Studies show that most of the days of dust during these two years start from late March and continue until early August. To compare the distribution of surface exposure lines in the studied area during normal days and when the dust existed, Figure 13 was plotted by Surfer software. The flow lines of the water temperature on 9 August 2008 showed that dust was not scattered in the sky (Figure 13A,a). The flow lines of the water temperature on 9 August 2009 show that horizontal vision is about 500–300 m (Figure 13B,b). The investigation of these two forms together shows that when dust is at a maximum in the sky, the surface temperature of the water has lower values.

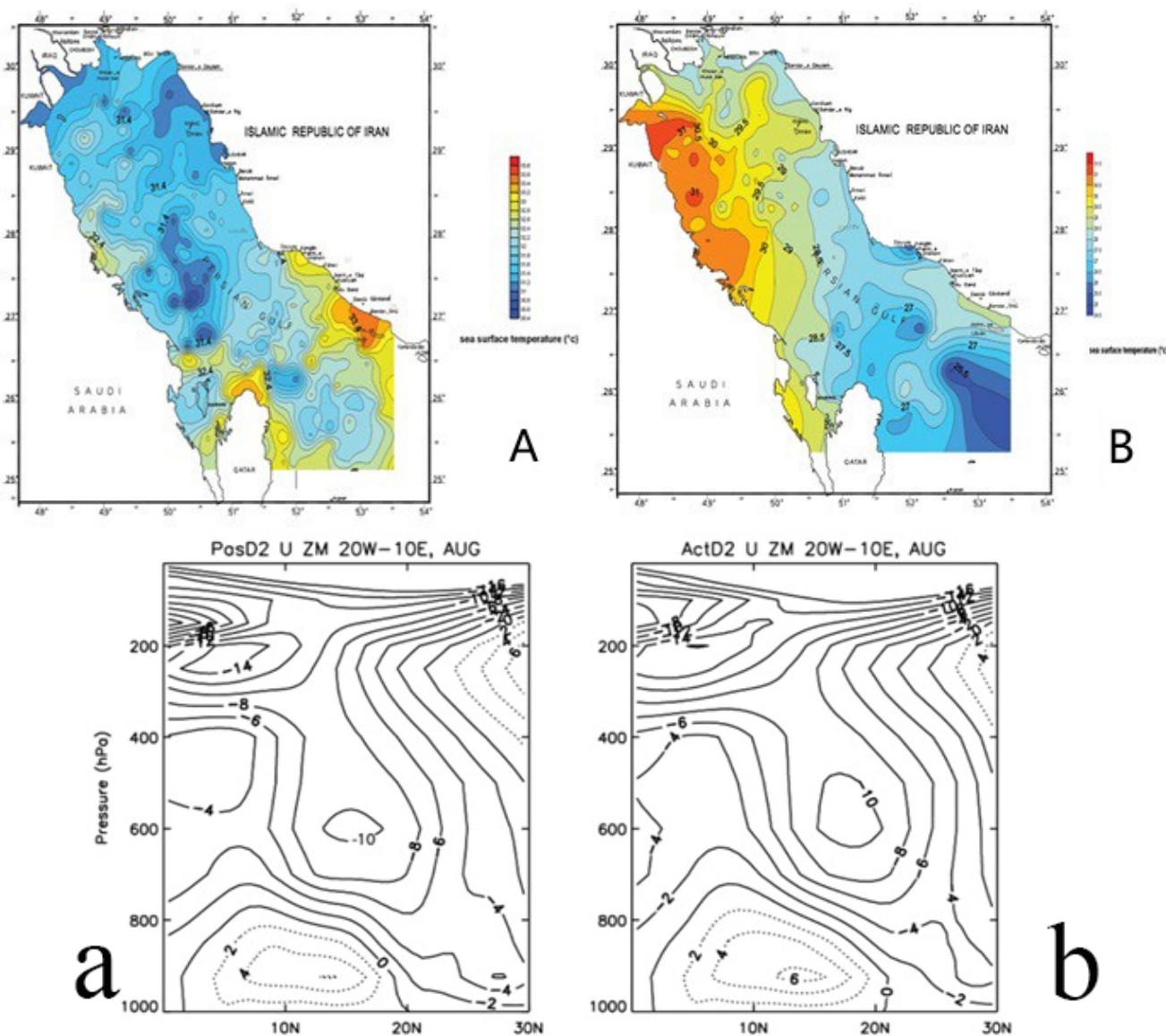

**Figure 13.** Surface water temperatures on August 9 in a year (**A**,**a**) 2008 that there was no dust and (**B**,**b**) 2009 which has the maximum dust and the dust concentration contours.

On the 7, 8, and 9 August 2008, where the dust was lasrge in the sky, the surface temperature of the water slightly decreased. At the northwest of the Persian Gulf, the surface temperature is lower, and the surface temperature increases toward the east.

The study of distribution of surface temperature lines in the studied area shows that the direction of isothermal lines in summer is hyponym to east and west. This means that in the summer, due to increase in air temperature, and evaporation and of course, maximum dust phenomena, the effects of surface temperature change due to surface flow and flow of rivers decreased. The investigation of surface water temperature maps in summer shows that the maximum amount of water temperature during neighboring waters with the shores of Bushehr province and in the central regions of the northern Persian Gulf section with temperatures of approximately 34 to 36 degrees is observed. The surface flows on the shores of Bushehr province are created due to the North West winds, and in the hot season of the year the intensity of these winds, and the result of the surface flows due to decrease in wind. The study of surface temperature maps in winter shows that warmer of water in neighboring areas appear with the Oman Sea, and the Strait of Hormuz and to the northwest of the Persian Gulf is reduced from water temperature. The cause of this is the

entry of warmer water from the Oman Sea to the Persian Gulf and the onscreen is clearly seen in the maps, with the entry of this water, neighboring waters with the Iranian coast achieved more temperature. In the spring, the temperature of the inlet water is raised from the Strait of Hormuz and reaches 29 °C. In a map that shows the northwest of the Persian Gulf, the temperature reduced, and it reaches about 23 °C in some areas. In this season, onscreen is visible in the surface area, the entry of Arvand river water has led to a decrease in the temperature in the northwest of Persian Gulf.

At the beginning of August 2009, the atmosphere was relatively calm and stability was achieved over the ruling area. However, in the middle of the first week of this month (5 August) in a domestic gradient of the land in the southern regions of the region, the extent of north currents intensified and caused mental dust and decreased horizontal views in this area. This situation was formed by the resonance of atmospheric unrest on Iraq and the intensity of wind speed, colonies of dust in this country, with the northwest currents of these pesky particles suspended in the air to the west and southwest of the country, including Bushehr province, and caused a severe reduction in air quality. The concentration of these particulate matters was due to the extent that on August 8, the horizontal views in Bushehr were reduced to 400 m and the flights were canceled on several consecutive days. In the second mid of the week, this situation was improved. In the south of the region, the southeast of the Indian Ocean systems, this resulted in invisibility in the day of the region, and the same mental the incidence of dust and reduced visibility in a cross-alternation. This situation is continuing in the south of the region almost in most days of this month. The temperature in this month has little fluctuations because the increase in moisture prevents severe temperature changes.

## 5. Conclusions

The main centers of the formation of dust phenomena in Iraq, and between Tigris and Euphrates (Bin Al-Nahrin), western and southern regions of Iraq, Kuwait, North-Saudi, and east of Jordan and South coast of the Persian Gulf, and western winds to the east have been transferring to Iran. The diameter of these very small soil particles starts from 10 μm, and in some cases, it reaches less than 5.2 μm.

Increasing dust particles in barley causes them to stabilize the Haze layer and these particles cannot climb to the higher layers of the atmosphere there adjacent to the surface of the ground and causes an increase in the air temperature. Increasing the air temperature of the sea level increases the temperature of water and this change increases the evaporation. Further evaporation causes the energy to be entered into the atmosphere in the form of latent heat. A part of the freed energy causes air warming, parts of which kinetic the air of energy and causes its movement and wind. Both factors increase temperature and wind creation resulting in increased evaporation. On the other hand, the rise in evaporation increases the amount and thickness of cloud cover, resulting in a decrease in the cloud. The short radiation has solar waves, thereby reducing the air temperature and also evaporation from the surface.

Changes in surface temperature of the water have an annual cycle. However, during the dust, the surface temperature of the water increased by increasing the air temperature, and then the surface temperature of the water reduced due to increase in evaporation. The reduction in surface water temperature in summer is due to the influx of dust and in autumn and winter it is due to reduction in sun radiation, reduction in air temperature, increasing wind speed, or the possibility of the occurrence of the weld because the composition of the wind sea-beach and the surface of the water is created by the Gulf in winter and spring.

During the occurrence of dust, the level of horizontal view is dramatically reduced. As a result of this, dust and pollutants are heavily dispersed in the air.

## 6. Recommendations for Future Works

The study of dust phenomenon in Iran and the Persian Gulf coastal regions is essential for human health, overall natural environment and human life. Therefore, the follow-

ing is recommended for a comprehensive study of the effect of mental dust on physical parameters of seawater and environment:

- Considering that the mental of the mist may have an effect on the Persian Gulf currents and coastal water movement physics and this can have the effective role in the region's climate, environment, fisheries, changing the shape of coastal areas and sea transport, it is necessary to investigate the effect of dust phenomenon on coastal flows and the pattern of marine currents;
- Statistics should be obtained and compared in serenity years and natural and non-normal factors and preventive agents on this phenomenon should be examined;
- The sediments of the remainder should be examined in the provinces affected by the origin of the study and the strategies for coping with this phenomenon should be determined;
- Due to the 24-h imaging of the Earth by the sensors in the space, the obtained results can be tested using the remote assay technique;
- An investigation of the effect of dust phenomenon on water balance surface or sea surface pressure, using the results of this study.

**Author Contributions:** Conceptualization M.T.A., K.L., R.O., T.S. and O.K.; Data curation, M.T.A., K.L., R.O., T.S. and O.K.; Investigation, M.T.A., K.L., R.O., T.S. and O.K.; Methodology, M.T.A., K.L., R.O., T.S. and O.K.; Soft-ware, M.T.A., K.L., R.O., T.S. and O.K.; Validation, M.T.A., K.L., R.O., T.S. and O.K.; writing—original draft, M.T.A., K.L., R.O., T.S. and O.K.; Writing—review & editing, M.T.A., K.L., R.O., T.S. and O.K. All authors have read and agreed to the published version of the manuscript.

**Funding:** This research received no external funding.

**Institutional Review Board Statement:** Not applicable.

**Informed Consent Statement:** Not applicable.

**Data Availability Statement:** Meteorological Organization of Iran.

**Conflicts of Interest:** The authors declare no conflict of interest.

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
