# Peer review of "The Effect of Dust Storm on Sea Surface Temperature in the Western Basin of Persian Gulf"

_standards, doi:10.3390/standards2030018_

Round 1

Reviewer 1 Report

Comments and Suggestions for Authors

The manuscript entitled "The Effect of Dust Storm on Sea Surface Temperature in the Western Basin of Persian Gulf"  investigates the effect of dust phenomenon on the surface temperature of water in the western Persian Gulf basin and presents the necessary predictions for this phenomenon. However, there are still some questions to be considered as follows:
1. The introduction of the research background is not detailed enough. The references in this part are too few and old, and the author may not fully browse the existing relevant studies.
2.  The author should clearly point out the purpose of this study in the last paragraph of the introduction. 
3. The figure of the study area should be shown in the materials and methods section.
4. In lines 161-163, "The R2 value was estimated to be 0.9 and shows that there is a strong correlation between……"  at what confidence level? 0.01 or 0.05?
5. The conclusion should be further condensed and simply point out the main findings of this study. 
6.  The figures in this paper are not clear enough. In addition, figure 4-figure 10 can be merged as needed. 
7. There are too few references and lack of research in recent five years.

Author Response

Thanks for your valuable comments. Please see the attached file for our responses.

Reviewer 2 Report

Comments and Suggestions for Authors

The manuscript requires extensive improvements in science and writing before being accepted for publication. It is hard to read and to follow.

what is the main objective of the paper? Where are the research outcomes? 

Authors should think carefully about how to communicate the issue, how to address it and how to present it in an easy well written way. All sections requires to be rewritten.

Some specific comments are listed below

(1) Line 39: "The deployment of Bushehr province between 27 ℃ and 14 minutes to 30 ℃ and 16 39 minutes..". I don't understand this sentence. Is the  temperature ? But the temperature range in degree C  should be in decimal unit.

(2) Line 41: Change "at the distances of the geographical widths" to "and its width"

(3) Line 43: "The development" should be "The extent"

(4) Line 46: "has a warm climate that is warm" should be "climate is warm"

(5) Line 48: "systems" should be "factors"

(6) Line 52-53: "During hot months of the year, especially during the summer of the low pressure.." should be "During the hot summer months, the low pressure ..."

(7) Line 63-64: "and the Low-Pressure Center tab penetration in these areas with winds, which because of north West of Iran," should be "and the low-pressure system penetration in North West of Iran,.."

(8) Line 67: "currents" should be "wind flows"

(9) Line 104 : Poor sentence "The dust of HYSPLIT is based on the algorithm for the release of dust storms to use desert land".

.. There are numerous bad sentences and wordings in the text. The authors need a native English speaker to help in grammar and  syntax correction in the manuscript.

(10) Line 134: In Material and method section, satellite data was mentioned but not used in here. The first paragraph should be in the Introduction

(11) Line 203: In the Discussion section, it is very difficult to follow. The graphs of the 2 years (2008 and 2009) should be plotted together for each of the wind speed, air temperature and sea surface temperature so that comparison can be made easily.

(12) Line 275: "scope of the wind" Should be range of the wind speed"

(13) Figure 16: The authors should provide dust concentration contour as well. The wind field should also be given.

(14) Line 437 : "the study of dust mental" What is dust mental ?

Author Response

(The authors gave the same response as above.)

Round 2

Reviewer 1 Report

Comments and Suggestions for Authors

Accept in present form. 

Author Response

Thanks for your positive response.

Reviewer 2 Report

Comments and Suggestions for Authors

My recommendation to the authors are as follows:

  • The introduction section is too long. it can be reduced without losing contents.
  • Remove repetition from the manuscript
  • The English writing can be improved. some sections still hard to read. Some specific comments:
  1. Line 4: “location” rather than “deployment”
  2. Line 107-113: These sentences are difficult to read and has serious English problem
  3. Line 147: change “dust” to “wind speed”
  4. The authors do not address our previous comment. Figure 4 and Figure 6 should be combined together as well as Figure 7 and 8. Missing Figure on wind direction for 2009 which should be combined with Figure 5.
  5. The authors should provide the plot of the dust concentration contours next to Figure 16 as I commented previously
  6. Line 466: Remove “Essentially”

Author Response

Thanks for your queries. Please see our responses in the attachment.

Round 3

Reviewer 2 Report

Comments and Suggestions for Authors

the paper is ready to be published. However, I suggest the following:

1- rewrite the paragraph located between lines 45 and 47

2- Section between 49-68 can be trimmed to highlight the important message

3- Remove dots from the wind graphs for clarity

4- some charts have labels cross over the borders

After these changes the paper should be published 

Author Response

Thank you very much for your suggestion. All the suggestions were considered.
